# A Simple and Provably Efficient Algorithm for Asynchronous Federated Contextual Linear Bandits

**Jiafan He**[*]
Department of Computer Science
University of California, Los Angeles
Los Angeles, CA 90095
jiafanhe19@ucla.edu

**Tianhao Wang**[*]
Department of Statistics and Data Science
Yale University
New Haven, CT 06511
tianhao.wang@yale.edu

**Yifei Min**[*]
Department of Statistics and Data Science
Yale University
New Haven, CT 06511
yifei.min@yale.edu

**Quanquan Gu**
Department of Computer Science
University of California, Los Angeles
Los Angeles, CA 90095
qgu@cs.ucla.edu

## Abstract

We study federated contextual linear bandits, where $M$ agents cooperate with each other to solve a global contextual linear bandit problem with the help of a central server. We consider the asynchronous setting, where all agents work independently and the communication between one agent and the server will not trigger other agents' communication. We propose a simple algorithm named `FedLinUCB` based on the principle of optimism. We prove that the regret of `FedLinUCB` is bounded by $\widetilde{O}(d\sqrt{\sum_{m=1}^{M} T_m})$ and the communication complexity is $\widetilde{O}(dM^2)$, where $d$ is the dimension of the contextual vector and $T_m$ is the total number of interactions with the environment by $m$-th agent. To the best of our knowledge, this is the first provably efficient algorithm that allows fully asynchronous communication for federated contextual linear bandits, while achieving the same regret guarantee as in the single-agent setting.

## 1 Introduction

Contextual linear bandit is a canonical model in sequential decision making with partial information feedback that has found vast applications in real-world domains such as recommendation systems (Li et al., 2010a,b; Gentile et al., 2014; Li et al., 2020), clinical trials (Wang, 1991; Durand et al., 2018) and economics (Jagadeesan et al., 2021; Li et al., 2022). Most existing works on contextual linear bandits focus on either the single-agent setting (Auer, 2002; Abe et al., 2003; Dani et al., 2008; Li et al., 2010a; Rusmevichientong and Tsitsiklis, 2010; Chu et al., 2011; Abbasi-Yadkori et al., 2011; Agrawal and Goyal, 2013) or multi-agent settings where communications between agents are instant and unrestricted (Cesa-Bianchi et al., 2013; Li et al., 2016; Wu et al., 2016; Li et al., 2021). Due to the increasing amount of data being distributed across a large number of local agents (e.g., clients, users, edge devices), federated learning (McMahan et al., 2017; Karimireddy et al., 2020) has become an emerging paradigm for distributed machine learning, where agents can jointly learn a global model without sharing their own localized data. This motivates the development of distributed/federated linear bandits (Wang et al., 2019; Huang et al., 2021; Li and Wang, 2022a), which enables a collection of agents to cooperate with each other to solve a global linear bandit

---

[*]Equal contribution.

36th Conference on Neural Information Processing Systems (NeurIPS 2022).

problem while enjoying performance guarantees comparable to those in the classical single-agent centralized setting.

However, most existing federated linear bandits algorithms are limited to the synchronous setting (Wang et al., 2019; Dubey and Pentland, 2020; Huang et al., 2021), where all the agents have to first upload their local data to the server upon the request of the server, and the agents will download the latest data from the server after all uploads are complete. This requires full participation of the agents and global synchronization mandated by the server, which is impractical in many real-world application scenarios. The only notable exception is Li and Wang (2022a), where an asynchronous federated linear bandit algorithm is proposed. Nevertheless, in their algorithm, the upload by one agent may trigger the download from the server to all other agents. Therefore, the communications between different agents and the server are not totally independent. Moreover, they make a stringent regularity assumption on the contexts, which basically requires the contexts to be stochastic rather than adversarial as in standard contextual linear bandits. That said, their theoretical proof is actually flawed as they ignored some unique challenges caused by asynchronous communication (see Appendix A.1 for details). Therefore, how to design a truly asynchronous contextual linear bandit algorithm remains an open problem.

In this work, we resolve the above open problem by proposing a simple algorithm for asynchronous federated contextual linear bandits over a star-shaped communication network. Our algorithm is based on the principle of optimism (Abbasi-Yadkori et al., 2011) and enjoys the following advantages: (i) Each agent can decide whether or not to participate in each round. Full participation is not required, thus it allows temporarily offline agents. This is much more flexible than existing algorithms for federated linear bandits in Wang et al. (2019); Dubey and Pentland (2020); Huang et al. (2021) where all agents are required to participate in each round; and (ii) the communication between each agent and the server is asynchronous and totally independent of other agents. There is no need of global synchronization or mandatory coordination by the server, in contrast to Li and Wang (2022a) where each agent might be asked by the server to download data. In particular, the communication between the agent and the server is triggered by a matrix determinant-based criterion that can be computed independently by each agent. Our algorithm design not only allows the agents to independently operate and synchronize with the server, but also ensures low communication complexity (i.e., total number of rounds of communication between agents and the server) and low switching cost (i.e., total number of local model updates for all agents) (Abbasi-Yadkori et al., 2011).

While being simple, our algorithm design introduces a challenge in the regret analysis. Since the order of the interaction between the agent and the environment is not fixed, standard martingale-based concentration inequality cannot be directly applied. Specifically, this challenge arises due to the mismatch between the partial data information collected by the central server and the true order of the data generated from the interaction with the environment, as is explained in detail in Section 5 and illustrated by Figure 1. We address this challenge by a novel proof technique, which first establishes the local concentration of each agent's data and then relates it to the "virtual" global concentration of all data via the determinant-based criterion. Based on this proof technique, we are able to obtain tight enough confidence bounds that lead to a nearly optimal regret. Moreover, our theoretical analysis relies only on minimal assumptions that are standard for contextual linear bandits, relaxing the strong assumptions made in Li and Wang (2022a).

**Main contributions.** Our contributions are highlighted as follows:

- We devise a simple algorithm named `FedLinUCB` that achieves near-optimal regret, low communication complexity and low switching cost simultaneously for asynchronous federated contextual linear bandits. In detail, we prove that our algorithms achieves a near-optimal $\widetilde{O}(d\sqrt{T})$ regret with merely $\widetilde{O}(dM^2)$ total communication complexity and $\widetilde{O}(dM^2)$ total switching cost. Here $M$ is the number of agents, $d$ is the dimension of the context and $T = \sum_{m=1}^{M} T_m$ is the total number of rounds with $T_m$ being the number of rounds that agent $m$ participates in. When degenerated to single-agent bandits, the regret of our algorithm matches the optimal regret $\widetilde{O}(d\sqrt{T})$ (Abbasi-Yadkori et al., 2011).

- We also prove an $\Omega(M/\log(T/M))$ lower bound for the communication complexity. Together with the $O(dM^2)$ upper bound of our algorithm, it suggests that there is only an $\widetilde{O}(dM)$ gap between the upper and lower bounds of the communication complexity.

- We identify the issue of ill-defined filtration caused by the unfixed order of interactions between agents and the environment, which is absent in previous synchronous or single-agent settings. We tackle this unique challenge by connecting the local concentration of each local agent's data and the global concentration of the aggregated data from all agents. We believe this proof technique is of independent interest for the analysis of other asynchronous bandit problems.

**Notation.** For any positive integer $n$, we denote the set $\{1, 2, \ldots, n\}$ by $[n]$. We use $\mathbf{I}$ to denote the $d \times d$ identity matrix. We use $O$ to hide universal constants and $\widetilde{O}$ to further hide poly-logarithmic terms. For any vector $\mathbf{x} \in \mathbb{R}^d$ and positive semi-definite $\mathbf{\Sigma} \in \mathbb{R}^{d \times d}$, we denote $\|\mathbf{x}\|_{\mathbf{\Sigma}} = \sqrt{\mathbf{x}^\top \mathbf{\Sigma} \mathbf{x}}$.

## 2 Related Work

We review related work on distributed/federated bandit algorithms stratified by the type of bandits: (1) multi-armed bandits, (2) stochastic linear bandits and (3) contextual linear bandits.

**Distributed/federated multi-armed bandits.** There is a vast literature on distributed/federated multi-armed bandits (MABs) (Liu and Zhao, 2010; Szorenyi et al., 2013; Landgren et al., 2016; Chakraborty et al., 2017; Landgren et al., 2018; Martínez-Rubio et al., 2019; Sankararaman et al., 2019; Wang et al., 2019, 2020; Zhu et al., 2021), to mention a few. However, none of these algorithms can be directly applied to linear bandits, needless to say contextual linear bandits with infinite decision sets.

**Distributed/federated stochastic linear bandits.** In distributed/federated stochastic linear bandits, the decision set is fixed across all the rounds $t \in [T]$ and all the agents $m \in [M]$. Wang et al. (2019) proposed the DELD algorithm for distributed stochastic linear bandits on both star-shaped network and P2P network. Huang et al. (2021) proposed an arm elimination-based algorithm called Fed-PE for federated stochastic linear bandits on the star-shaped network. Both algorithms are in the synchronous setting and require full participation of the agents upon the server's request.

**Distributed/federated contextual linear bandits.** The contextual linear bandit is more general and challenging than stochastic linear bandits, because the decision sets can vary for each $t$ and $m$. In this setting, Korda et al. (2016) considered a P2P network and proposed the DCB algorithm based on the OFUL algorithm in Abbasi-Yadkori et al. (2011). Wang et al. (2019) considered both star-shaped and P2P communication networks and achieved the near-optimal $\widetilde{O}(d\sqrt{T})$ regret in the synchronous setting.[2] Dubey and Pentland (2020) further introduced the differential privacy guarantee into the setting of Wang et al. (2019). Li and Wang (2022b) extended distributed contextual linear bandits to generalized linear bandits (Filippi et al., 2010; Jun et al., 2017) in the synchronous setting. Li and Wang (2022a) proposed the first asynchronous algorithm for federated contextual linear bandits with the star-shaped graph and achieved a near-optimal $\widetilde{O}(d\sqrt{T})$ regret. However, their setting is different from ours in two aspects: (1) the upload triggered by an agent will lead the server to trigger download possibly for all the agents in their setting. In contrast, the upload triggered by an agent will only lead to download to the same agent in our setting; (2) their regret guarantee relies on a stringent regularity assumption on the contexts, which basically requires the contexts to be stochastic. As a comparison, the contexts in our setting can be even adversarial, which is exactly the standard setting of contextual linear bandits (Abbasi-Yadkori et al., 2011; Li et al., 2019). This difference in the setting makes our algorithm a truly asynchronous contextual linear bandit algorithm but also makes our regret analysis more challenging.

For better comparison, we compare our work with the most related contextual linear bandit algorithms in Table 1.

## 3 Preliminaries

**Federated contextual linear bandits.** We consider the federated contextual linear bandits as follows: At each round $t \in [T]$, an arbitrary agent $m_t \in [M]$ is active for participation. This agent receives a decision set $D_t \subset \mathbb{R}^d$, picks an action $\mathbf{x}_t \in D_t$, and receives a random reward $r_t$. We assume that

---

[2]In the original paper of Wang et al. (2019), the regret bound is expressed as $\widetilde{O}(d\sqrt{MT})$. The $T$ in their paper is equivalent to the $T_m$ in ours, so their $d\sqrt{MT}$ should be understood as $d\sqrt{T}$ under our notation.

| Setting | Algorithm | Regret | Communication | Low-switching | No extra assump. on contexts | Allow free participation |
|---------|-----------|--------|---------------|---------------|------------------------------|--------------------------|
| Single-agent | OFUL (Abbasi-Yadkori et al., 2011) | $d\sqrt{T\log T}$ | N/A | ✓ | ✓ | ✗ |
| Federated (Sync.) | DisLinUCB (Wang et al., 2019) | $d\sqrt{T}\log^2 T$ | $dM^{3/2}$ | ✗ | ✓ | ✗ |
| Federated (Async.) | Async-LinUCB (Li and Wang, 2022a) | $d\sqrt{T}\log T$ | $dM^2\log T$ | ✗ | ✗ | ✗ |
| Federated (Async.) | FedLinUCB (Our Algorithm 1) | $d\sqrt{T}\log T$ | $dM^2\log T$ | ✓ | ✓ | ✓ |

Table 1: Comparison of our result with baseline approaches for contextual linear bandits. Our result achieves near-optimal regret under low communication complexity. Here $d$ is the dimension of the context, $M$ is the number of agents, and $T = \sum_{m=1}^{M} T_m$ with each $T_m$ being the number of rounds that agent $m$ participates in.

the reward $r_t$ satisfies $r_t = \langle \mathbf{x}_t, \boldsymbol{\theta}^* \rangle + \eta_t$ for all $t \in [T]$, where $\eta_t$ is conditionally independent of $\mathbf{x}_t$ given $\mathbf{x}_{1:t-1}, m_{1:t}, r_{1:t-1}$. More specifically, we make the following assumption on $\eta_t, \boldsymbol{\theta}^*$ and $D_t$, which is a standard assumption in the contextual linear bandit literature (Abbasi-Yadkori et al., 2011; Wang et al., 2019; Dubey and Pentland, 2020).

**Assumption 3.1.** The noise $\eta_t$ is $R$-sub-Gaussian conditioning on $\mathbf{x}_{1:t}, m_{1:t}$ and $r_{1:t-1}$, i.e.,

$$\mathbb{E}\left[e^{\lambda \eta_t} \mid \mathbf{x}_{1:t}, m_{1:t}, r_{1:t-1}\right] \leq \exp(R^2\lambda^2/2), \quad \text{for any } \lambda \in \mathbb{R}.$$

We also assume that $\|\boldsymbol{\theta}^*\|_2 \leq S$ and $\|\mathbf{x}\|_2 \leq L$ for all action $\mathbf{x} \in \mathcal{D}_t$, for all $t \in [T]$.

Notably, we assume $m_t$ can be arbitrary for all $t$, which basically says that each agent can decide whether and when to participate or not.[3] Our setting is more general than the synchronous setting in Wang et al. (2019); Dubey and Pentland (2020), which requires a round-robin participation of all agents.

**Learning objective.** The goal of the agents is to collaboratively minimize the cumulative regret defined as

$$\text{Regret}(T) := \sum_{t=1}^{T} \left( \max_{\mathbf{x} \in D_t} \langle \mathbf{x}, \boldsymbol{\theta}^* \rangle - \langle \mathbf{x}_t, \boldsymbol{\theta}^* \rangle \right) = \sum_{t=1}^{T} \langle \mathbf{x}_t^* - \mathbf{x}_t, \boldsymbol{\theta}^* \rangle. \tag{3.1}$$

To achieve such a goal, we allow the agents to collaborate via communication through the central server. Below we will explain the details of the communication model.

**Communication model.** We consider a star-shaped communication network (Wang et al., 2019; Dubey and Pentland, 2020) consisting of a central server and $M$ agents, where each agent can communicate with the server by uploading and downloading data. However, any pair of agents cannot directly communicate with each other. We define the communication complexity as the total number of communication rounds between agents and the server (counting both the uploads and downloads) (Wang et al., 2019; Dubey and Pentland, 2020; Li and Wang, 2022a). For simplicity, we assume that there is no latency in the communication channel.

We consider the asynchronous setting, where the communication protocol satisfies: (1) each agent can decide whether or not to participate in each round. Full participation is not required, which allows temporarily offline agents; and (2) the communication between each agent and the server is asynchronous and independent of other agents without mandatory download required by the server.

**Switching cost.** The notion of switching cost in online learning and bandits refers to the number of times the agent switches its policy (i.e., decision rule) (Kalai and Vempala, 2005; Abbasi-Yadkori et al., 2011; Dekel et al., 2014; Ruan et al., 2021). In the context of linear bandits, it corresponds to the number of times the agent updates its policy of selecting an action from the decision set (Abbasi-Yadkori et al., 2011). Algorithms with low switching cost are preferred in practice since each policy switching might cause additional computational overhead.

---

[3]Without loss of generality, we can assume that it cannot happen that more than one agent participate at the same time. Therefore, there is always a valid order of participation indexed by $t \in [T]$.

---

**Algorithm 1** Federated linear UCB (`FedlinUCB`)

---

1: Initialize $\boldsymbol{\Sigma}_{m,1} = \boldsymbol{\Sigma}_1^{\text{ser}} = \lambda \mathbf{I}$, $\widehat{\boldsymbol{\theta}}_{m,1} = 0$, $\mathbf{b}_{m,0}^{\text{loc}} = 0$ and $\boldsymbol{\Sigma}_{m,0}^{\text{loc}} = 0$ for all $m \in [M]$
2: **for** round $t = 1, 2, \ldots, T$ **do**
3:      Agent $m_t$ is active
4:      Receive $D_t$ from the environment
5:      Select $\mathbf{x}_t \leftarrow \operatorname{argmax}_{\mathbf{x} \in D_t} \langle \widehat{\boldsymbol{\theta}}_{m_t,t}, \mathbf{x} \rangle + \beta \|\mathbf{x}\|_{\boldsymbol{\Sigma}_{m_t,t}^{-1}}$            `/* Optimistic decision */`
6:      Receive $r_t$ from environment
7:      $\boldsymbol{\Sigma}_{m_t,t}^{\text{loc}} \leftarrow \boldsymbol{\Sigma}_{m_t,t-1}^{\text{loc}} + \mathbf{x}_t \mathbf{x}_t^\top$,     $\mathbf{b}_{m_t,t}^{\text{loc}} \leftarrow \mathbf{b}_{m_t,t-1}^{\text{loc}} + r_t \mathbf{x}_t$          `/* Local update */`
8:      **if** $\det(\boldsymbol{\Sigma}_{m_t,t} + \boldsymbol{\Sigma}_{m_t,t}^{\text{loc}}) > (1+\alpha)\det(\boldsymbol{\Sigma}_{m_t,t})$ **then**
9:          Agent $m_t$ sends $\boldsymbol{\Sigma}_{m_t,t}^{\text{loc}}$ and $\mathbf{b}_{m_t,t}^{\text{loc}}$ to server                   `/* Upload */`
10:        $\boldsymbol{\Sigma}_t^{\text{ser}} \leftarrow \boldsymbol{\Sigma}_t^{\text{ser}} + \boldsymbol{\Sigma}_{m_t,t}^{\text{loc}}$,     $\mathbf{b}_t^{\text{ser}} \leftarrow \mathbf{b}_t^{\text{ser}} + \mathbf{b}_{m_t,t}^{\text{loc}}$        `/* Global update */`
11:        $\boldsymbol{\Sigma}_{m_t,t}^{\text{loc}} \leftarrow 0$,     $\mathbf{b}_{m_t,t}^{\text{loc}} \leftarrow 0$
12:        Server sends $\boldsymbol{\Sigma}_t^{\text{ser}}$ and $\mathbf{b}_t^{\text{ser}}$ back to agent $m_t$                `/* Download */`
13:        $\boldsymbol{\Sigma}_{m_t,t+1} \leftarrow \boldsymbol{\Sigma}_t^{\text{ser}}$,     $\mathbf{b}_{m_t,t+1} \leftarrow \mathbf{b}_t^{\text{ser}}$
14:        $\widehat{\boldsymbol{\theta}}_{m_t,t+1} \leftarrow \boldsymbol{\Sigma}_{m_t,t+1}^{-1} \mathbf{b}_{m_t,t+1}$                  `/* Compute estimate */`
15:      **else**
16:        $\boldsymbol{\Sigma}_{m_t,t+1} \leftarrow \boldsymbol{\Sigma}_{m,t}$,     $\mathbf{b}_{m_t,t+1} \leftarrow \mathbf{b}_{m,t}$,     $\widehat{\boldsymbol{\theta}}_{m_t,t+1} \leftarrow \widehat{\boldsymbol{\theta}}_{m_t,t}$
17:      **end if**
18:      **for** other inactive agent $m \in [M] \setminus \{m_t\}$ **do**
19:        $\boldsymbol{\Sigma}_{m,t+1} \leftarrow \boldsymbol{\Sigma}_{m,t}$,     $\mathbf{b}_{m,t+1} \leftarrow \mathbf{b}_{m,t}$,     $\widehat{\boldsymbol{\theta}}_{m,t+1} \leftarrow \widehat{\boldsymbol{\theta}}_{m,t}$
20:      **end for**
21: **end for**

---

## 4 The Proposed Algorithm

We propose a simple algorithm based on the principle of optimism that enables collaboration among agents through asynchronous communications with the central server. The main algorithm is displayed in Algorithm 1. For clarity, we first summarize the related notations in Table 2.

| Notation | Meaning |
|---|---|
| $\widehat{\boldsymbol{\theta}}_{m,t}$ | estimate of $\boldsymbol{\theta}^*$ |
| $\boldsymbol{\Sigma}_{m,t}, \mathbf{b}_{m,t}$ | data used to compute $\widehat{\boldsymbol{\theta}}_{m,t}$ |
| $\boldsymbol{\Sigma}_{m,t}^{\text{loc}}, \mathbf{b}_{m,t}^{\text{loc}}$ | local data for agent $m$ |
| $\boldsymbol{\Sigma}_{m,t}^{\text{ser}}, \mathbf{b}_{m,t}^{\text{ser}}$ | data stored at the server |

Table 2: Notations used in Algorithm 1.

Specifically, in each round $t \in [T]$, agent $m_t$ participates and interacts with the environment (Line 3). The environment specifies the decision set $D_t$ (Line 4), and the agent $m_t$ selects the action based on its current optimisitic estimate of the reward (Line 5). Here the bonus term $\beta \|\mathbf{x}\|_{\boldsymbol{\Sigma}_{m_t,t}^{-1}}$ reflects the uncertainty of the estimated reward $\langle \widehat{\boldsymbol{\theta}}_{m_t,t}, \mathbf{x} \rangle$ and encourages exploration. After receiving the true reward $r_t$ from the environment, agent $m_t$ then updates its local data (Line 7).

The key component of the algorithm is the matrix determinant-based criterion (Line 8), which evaluates the information accumulated in current local data. If the criterion is satisfied, it suggests that the local data would help significantly reduce the uncertainty of estimating the model $\boldsymbol{\theta}^*$. Therefore, agent $m_t$ will share its progress by uploading the local data to the server (Line 9) so that it can benefit other agents. Then the server updates the global data accordingly (Line 10). Afterwards, agent $m_t$ downloads the latest global data from the server (Line 12) and updates its local data and model (Line 13-14). If the criterion in Line 8 is not met, then the communication between the agent and the server will not be triggered, and the local data remains local and unshared for agent $m_t$ (Line 16). Finally, all the other inactive agents remain unchanged (Line 19).

Note that in Algorithm 1, the communication between the agent and the server (Line 9 and 12) involves only the active agent in that round, which is completely independent of other agents. This is in sharp contrast to existing algorithms. For example, in the main algorithm in Li and Wang (2022a), upload by any agent may trigger other agents to download the latest data, while our algorithm does not mandate this. On the other hand, many existing algorithms for multi-agent settings (e.g., Wang

[et al. (2019)](#)) require all agents to interact with the environment in each round, which essentially require full participation of all the agents.

The determinant-based criterion in Line 8 has been a long-standing design trick in contextual linear bandits that can help address the issue of unknown time horizon and reduce the switching cost (Abbasi-Yadkori et al., 2011; Ruan et al., 2021). For multi-agent bandits, such a criterion has also been used to control the communications complexity (Wang et al., 2019; Dubey and Pentland, 2020; Li and Wang, 2022a). This is because the need for policy switching or communication essentially reflects the same fact: enough information has been collected and it is time to update the (local) model. Indeed, achieving low communication complexity and low switching cost are unified in our `FedlinUCB` algorithm in the sense that the communication complexity is exactly twice the switching cost. Furthermore, using lazy update makes our algorithm amenable for analysis, which will be clear later in Section 6. In addition, we leave $\alpha > 0$ as a tuning parameter as it controls the trade-off between the regret and the communication complexity.

## 5 Theoretical Results

We now present our main result on the theoretical guarantee of Algorithm 1.

**Theorem 5.1.** Under Assumption 3.1 for Algorithm 1, if we set the confidence radius $\beta = \sqrt{\lambda}S + (\sqrt{1 + M\alpha} + M\sqrt{2\alpha})\Big(R\sqrt{d\log\Big(\big(1 + TL^2/(\min(\alpha,1)\lambda)\big)/\delta\Big)} + \sqrt{\lambda}S\Big)$, then with probability at least $1 - \delta$, the regret in the first $T$ rounds can be upper bounded by

$$\text{Regret}(T) \leq 2dSLM\log(1 + TL^2/\lambda) + 2\sqrt{2(1 + M\alpha)}\beta\sqrt{2dT\log(1 + TL^2/\lambda)}.$$

Moreover, the communication complexity and switching cost are both bounded by $2\log 2 \cdot d(M + 1/\alpha)\log(1 + TL^2/(\lambda d))$.

**Remark 5.2.** Theorem 5.1 suggests that if we set the parameters $\alpha = 1/M^2$ and $\lambda = 1/S^2$ in Algorithm 1, then its regret is bounded by $\widetilde{O}(Rd\sqrt{T})$ and the corresponding communication complexity and switching cost are both bounded by $\widetilde{O}(dM^2)$. This choice of parameters yields the regret bound and the communication complexity presented in Table 1.

As a complement, we also provide a lower bound for the communication complexity as stated in the following theorem. See Appendix D for the proof.

**Theorem 5.3.** For any algorithm **Alg** with expected communication complexity less than $O(M/\log(T/M))$, there exist a linear bandit instance with $R = L = S = 1$ such that for $T \geq Md$, the expected regret for algorithm **Alg** is at least $\Omega(d\sqrt{MT})$.

**Remark 5.4.** Suppose each agent runs the `OFUL` algorithm (Abbasi-Yadkori et al., 2011) separately, then each agent $m \in [M]$ admits an $\widetilde{O}(d\sqrt{T_m})$ regret, where $T_m$ is the number of rounds that agent $m$ participates in. Thus the total regret of $M$ agents is upper bounded by $\sum_{m=1}^{M}\widetilde{O}(\sqrt{T_m}) = \widetilde{O}(d\sqrt{MT})$. Theorem 5.3 implies that for any algorithm **Alg**, if its communication complexity is less than $O(M/\log(T/M))$, then its regret cannot be better than naively running $M$ independent `OFUL` algorithms. In other words, Theorem 5.3 suggests that in order to improve the performance through collaboration, an $\Omega(M)$ communication complexity is necessary.

## 6 Overview of the Proof

When analyzing the performance of `FedLinUCB`, we face a unique challenge caused by the asynchronous communication, as illustrated in Figure 1. Here $(\mathbf{x}_{m,t}, \eta_{m,t})$ denotes the decision and the noise for agent $m$ in its own $t$-th round. Specifically, in the synchronous setting, the filtration is generated by all the data collected by all agents, i.e., $\mathcal{F}_5 = \sigma\{\mathbf{x}_{m,t}, \eta_{m,t}\}_{t=1,m=1}^{5,5}$, as marked by the green dashed rectangle. This kind of filtration is well-defined since all agents share their data with each other at the end of each round. In sharp contrast, in our asynchronous setting, the data at the server can be generated by an irregular set of data from the agents, as marked by the blue rectangles. Such data pattern can be arbitrary and depends on the data collected in all previous rounds, which prevents us from defining a fixed filtration as we can do in the synchronous setting. Since the

application of standard martingale concentration inequalities relies on the well-defined filtration, they cannot be directly applied to our asynchronous setting.

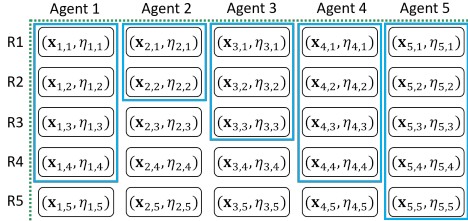

Figure 1: Illustration of ill-defined filtration.

To circumvent the above issue, we need to analyze the concentration property of the local data for each agent and then relate it to the concentration of the global data, so that we can control the sum of the bonuses and hence the regret. This requires a careful quantitative comparison of the local and global data covariance matrices, which is enabled by our design of determinant-based criterion. The details will be further explained in Section 6.2. In the next subsection, we present the key ingredients of the proof of Theorem 5.1.

**Remark 6.1.** Recall the notations in Table 2, where the values of those matrices and vectors might change within each round. To eliminate the possible confusion, from now on we follow the convention that all matrices and vectors in the analysis correspond to their values at the end of each round in Algorithm 1.

## 6.1 Analysis for communication complexity and switching cost

We first analyze the communication complexity and switching cost of Algorithm 1. For each $i \geq 0$, we define

$$\tau_i = \min\{t \in [T] \mid \det(\mathbf{\Sigma}_t^{\text{ser}}) \geq 2^i \lambda^d\}. \tag{6.1}$$

We divide the set of all rounds into epochs $\{\tau_i, \tau_i + 1, \ldots, \tau_{i+1} - 1\}$ for each $i \geq 0$. Then the communication complexity *within each epoch* can be bounded using the following lemma.

**Lemma 6.2.** Under the setting of Theorem 5.1, for each epoch from round $\tau_i$ to round $\tau_{i+1} - 1$, the number of communications in this epoch is upper bounded by $2(M + 1/\alpha)$.

*Proof of Theorem 5.1: communication complexity and switching cost.* It suffices to bound the number of epochs. By Assumption 3.1, we have $\|\mathbf{x}_t\|_2 \leq L$ for all $t \in [T]$. Since $\mathbf{\Sigma}_T^{\text{ser}}$ is positive definite, by the inequality of arithmetic and geometric means, we have

$$\det(\mathbf{\Sigma}_T^{\text{ser}}) \leq \left( \frac{\text{tr}(\mathbf{\Sigma}_T^{\text{ser}})}{d} \right)^d \leq \left( \frac{1}{d} \text{tr} \left( \lambda \mathbf{I} + \sum_{t=1}^{T} \mathbf{x}_t \mathbf{x}_t^\top \right) \right)^d$$

$$= \left( \lambda + \frac{1}{d} \sum_{t=1}^{T} \|\mathbf{x}_t\|_2^2 \right)^d \leq \lambda^d \left( 1 + \frac{TL^2}{\lambda d} \right)^d.$$

Then recalling the definition of epochs based on (6.1), we have

$$\max\{i \geq 0 \mid \tau_i \neq \emptyset\} = \log_2 \frac{\det(\mathbf{\Sigma}_T^{\text{ser}})}{\lambda^d} \leq \log 2 \cdot d \log \left( 1 + \frac{TL^2}{\lambda d} \right).$$

Therefore, the total number of epochs is bounded by $\log 2 \cdot d \log(1 + TL^2/(\lambda d))$. Now applying Lemma 6.2, the total communication complexity is bounded by $2 \log 2 \cdot d(M + 1/\alpha) \log(1 + TL^2/(\lambda d))$. Note that in Algorithm 1, each agent only switch its policy after communicating with the server, so the switching cost is exactly equal to half of the communication complexity. This finishes the proof for the claim on communication complexity and switching cost in Theorem 5.1. □

## 6.2 Analysis for regret upper bound

The regret analysis for Theorem 5.1 is much more involved, and it relies on a series of intermediate lemmas establishing the concentration.

**Total information.** We define the following auxiliary matrices and vectors that contain all the information up to round $t$:

$$\mathbf{\Sigma}_t^{\text{all}} = \lambda \mathbf{I} + \sum_{i=1}^{t} \mathbf{x}_i \mathbf{x}_i^\top, \qquad \mathbf{b}_t^{\text{all}} = \sum_{i=1}^{t} r_i \mathbf{x}_i, \qquad \mathbf{u}_t^{\text{all}} = \sum_{i=1}^{t} \eta_i \mathbf{x}_i, \tag{6.2}$$

where $\eta_i := r_i - \langle \mathbf{x}_i, \boldsymbol{\theta}^* \rangle$ is a $R$-sub-Gaussian noise by Assumption 3.1. In our setting, $\boldsymbol{\Sigma}_t^{\text{all}}, \mathbf{b}_t^{\text{all}}, \mathbf{u}_t^{\text{all}}$ are not accessible by the agents due to asynchronous communication, and we only use them to facilitate the analysis. With this notation, we can further define the following omnipotent estimate:

$$\widehat{\boldsymbol{\theta}}_t^{\text{all}} = (\boldsymbol{\Sigma}_t^{\text{all}})^{-1} \mathbf{b}_t^{\text{all}}. \tag{6.3}$$

As a direct application of the self-normalized martingale concentration inequality (Abbasi-Yadkori et al., 2011), we have the following global confidence bound due to the concentration of $\boldsymbol{\Sigma}_t^{\text{all}}$ and $\mathbf{b}_t^{\text{all}}$.

**Lemma 6.3** (Global confidence bound; Theorem 2, Abbasi-Yadkori et al. 2011). With probability at least $1 - \delta$, for each round $t \in [T]$, the estimate $\widehat{\boldsymbol{\theta}}_t^{\text{all}}$ in (6.3) satisfies

$$\|\widehat{\boldsymbol{\theta}}_t^{\text{all}} - \boldsymbol{\theta}^*\|_{\boldsymbol{\Sigma}_t^{\text{all}}} \le R\sqrt{d \log\left((1 + TL^2/\lambda)/\delta\right)} + \sqrt{\lambda} S.$$

**Per-agent information.** Next, for each agent $m \in [M]$, we denote the rounds when agent $m$ communicate with the server (i.e., upload and download data) as $\{t_{m,1}, t_{m,2}, ...\}$. For simplicity, *at the end of round $t$*, we denote by $N_m(t)$ the last round when agent $m$ communicated with the server (so if agent $m$ communicated with the server in round $t$, then $N_m(t) = t$). With this notation, for each round $t$ and agent $m \in [M]$, the data that has been uploaded by agent $m$ is then[4]

$$\boldsymbol{\Sigma}_{m,t}^{\text{up}} = \sum_{j=1, m_j=m}^{N_m(t)} \mathbf{x}_j \mathbf{x}_j^\top, \qquad \mathbf{u}_{m,t}^{\text{up}} = \sum_{j=1, m_j=m}^{N_m(t)} \mathbf{x}_j \eta_j.$$

Correspondingly, the local data that has not been uploaded to the server is

$$\boldsymbol{\Sigma}_{m,t}^{\text{loc}} = \sum_{j=N_m(t)+1, m_j=m}^{t} \mathbf{x}_j \mathbf{x}_j^\top, \qquad \mathbf{u}_{m,t}^{\text{loc}} = \sum_{j=N_m(t)+1, m_j=m}^{t} \mathbf{x}_j \eta_j.$$

Again, applying the self-normalized martingale concentration (Abbasi-Yadkori et al., 2011) together with a union bound, we can get the per-agent local concentration.

**Lemma 6.4** (Local concentration). Under the setting of Theorem 5.1, with probability at least $1 - \delta$, for each round $t \in [T]$ and each agent $m \in [M]$, it holds that

$$\left\| \left(\alpha\lambda\mathbf{I} + \boldsymbol{\Sigma}_{m,t+1}^{\text{loc}}\right)^{-1} \mathbf{u}_{m,t}^{\text{loc}} \right\|_{\alpha\lambda\mathbf{I} + \boldsymbol{\Sigma}_{m,t}^{\text{loc}}} \le R\sqrt{d \log\left(\left(1 + TL^2/(\alpha\lambda)\right)/\delta\right)} + \sqrt{\lambda} S.$$

Moreover, based on our determinant-based communication criterion, we have the following lemma describing the quantitative relationship among the local data, uploaded data and global data.

**Lemma 6.5** (Covariance comparison). Under the setting of Theorem 5.1, it holds that

$$\lambda\mathbf{I} + \sum_{m'=1}^{M} \boldsymbol{\Sigma}_{m',t}^{\text{up}} \succeq \frac{1}{\alpha} \boldsymbol{\Sigma}_{m,t}^{\text{loc}} \tag{6.4}$$

for each agent $m \in [M]$. Moreover, for any $1 \le t_1 < t_2 \le T$, if agent $m$ is the only active agent from round $t_1$ to $t_2 - 1$ and agent $m$ only communicates with the server at round $t_1$, then for all $t_1 + 1 \le t \le t_2$, it further holds that

$$\boldsymbol{\Sigma}_{m,t} \succeq \frac{1}{1 + M\alpha} \boldsymbol{\Sigma}_t^{\text{all}}. \tag{6.5}$$

Combining the above results, we obtain the local confidence bound, which then leads to the per-round regret in each round, as summarized in the following lemma.

**Lemma 6.6** (Local confidence bound and per-round regret). Under the setting of Theorem 5.1, with probability at least $1 - \delta$, for each $t \in [T]$, the estimate $\widehat{\boldsymbol{\theta}}_{m,t+1}$ satisfies that $\|\boldsymbol{\theta}^* - \widehat{\boldsymbol{\theta}}_{m,t+1}\|_{\boldsymbol{\Sigma}_{m,t+1}} \le \beta$ for all $m \in [M]$. Consequently, for each round $t \in [T]$, the regret in round $t$ satisfies

$$\Delta_t = \max_{\mathbf{x} \in \mathcal{D}_t} \langle \boldsymbol{\theta}^*, \mathbf{x} \rangle - \langle \boldsymbol{\theta}^*, \mathbf{x}_t \rangle \le 2\beta \sqrt{\mathbf{x}_t^\top \boldsymbol{\Sigma}_{m_t,t}^{-1} \mathbf{x}_t}.$$

---

[4]Strictly speaking, the uploaded data only consists of $\boldsymbol{\Sigma}_{m,t}^{\text{up}}$ and $\mathbf{b}_{m,t}^{\text{up}}$, and here we introduce $\mathbf{u}_{m,t}^{\text{up}}$ and $\mathbf{u}_{m,t}^{\text{loc}}$ solely for the purpose of analysis.

Now, we are ready to prove the regret bound in Theorem 5.1.

*Proof of Theorem 5.1: regret.* First, according to Lemma 6.6, the regret in first $T$ round can be decomposed and upper bounded by

$$\text{Regret}(T) = \sum_{t=1}^{T} \langle \boldsymbol{\theta}^*, \mathbf{x}_t^* - \mathbf{x}_t \rangle \leq \sum_{t=1}^{T} 2\beta \|\mathbf{x}_t\|_{\boldsymbol{\Sigma}_{m_t,t}^{-1}}.$$

Now, we only need to control the summation of the upper confidence bonus term $2\beta \|\mathbf{x}_t\|_{\boldsymbol{\Sigma}_{m_t,t}^{-1}}$ and we focus on the agent-action sequence $\{(m_t, \mathbf{x}_t)\}_{t=1}^{T}$. Notice that if an agent $m$ communicate with the server at time $t_1$ and $t_2$, then the order of actions between round $t_1$ and $t_2$ will not effect the covariance matrix for agent $m$. In addition, since agent $m$ does not upload new data between round $t_1$ and $t_2$, the order of actions from agent $m$ will also not affect other agents' covariance matrix. Thus, without effect the covariance matrix and the corresponding upper confidence bonus, we can always reorder the sequence of active agents such that each agent communicates with the server and stays active until the next agent kicks in to communicate with the server. Such reordering is valid according to the communication protocol as each agent has only local updates between communications with the server.

More specifically, we assume that the sequence of round that the active agent communicates with server is $0 = t_0 < t_1 < t_2 < \cdots < t_N = T + 1$[5], and from round $t_i + 1$ to $t_{i+1} - 1$ there is only one agent active, that is, $m_{t_i} = m_{t_i+1} = \cdots = m_{t_{i+1}-1}$. Then we apply Lemma 6.5 and get that the bonus term for each round $t_i < t < t_{i+1}$ can be controlled by

$$2\beta \|\mathbf{x}_t\|_{(\boldsymbol{\Sigma}_{m_t,t})^{-1}} \leq 2\beta \sqrt{1 + M\alpha} \|\mathbf{x}_t\|_{(\boldsymbol{\Sigma}_t^{\text{all}})^{-1}}. \tag{6.6}$$

In addition, to control the bonus term for rounds $\{t_i\}_{i=1}^{N}$, we define $T_i = \min\{t \in [T] \mid \det(\boldsymbol{\Sigma}_t^{\text{all}}) \geq 2^i \lambda^d\}$. For each time interval from $T_i$ to $T_{i+1}$, if an agent $m$ communicate with the server more than once, e.g., agent $m$ communicates with the server at round $T_{i,1}$ and $T_{i,2}$ such that $T_i \leq T_{i,1} < T_{i,2} < T_{i+1}$, then for the latter round $T_{i,2}$, the bonus term can be controlled by

$$2\beta \|\mathbf{x}_{T_{i,2}}\|_{(\boldsymbol{\Sigma}_{m,T_{i,2}})^{-1}} \leq 2\beta \sqrt{1 + M\alpha} \|\mathbf{x}_{T_{i,2}}\|_{(\boldsymbol{\Sigma}_{T_{i,1}}^{\text{all}})^{-1}}$$
$$\leq 2\beta \sqrt{2(1 + M\alpha)} \|\mathbf{x}_{T_{i,2}}\|_{(\boldsymbol{\Sigma}_{T_{i+1}-1}^{\text{all}})^{-1}}$$
$$\leq 2\beta \sqrt{2(1 + M\alpha)} \|\mathbf{x}_{T_{i,2}}\|_{(\boldsymbol{\Sigma}_{T_{i,2}}^{\text{all}})^{-1}}, \tag{6.7}$$

where the first inequality holds due Lemma 6.5 and agent $m$ communicate with the server at the previous round $T_{i,1}$, the second inequality holds due to Lemma E.4 with the fact that $\det \boldsymbol{\Sigma}_{T_{i+1}-1}^{\text{all}} / \det(\boldsymbol{\Sigma}_{T_{i,1}}^{\text{all}}) \leq 2^{i+1}\lambda^d / (2^i\lambda^d) = 2$, and the last inequality holds due to the fact that $\boldsymbol{\Sigma}_{T_{i+1}-1}^{\text{all}} \succeq \boldsymbol{\Sigma}_{T_{i,2}}^{\text{all}}$. On the other hand, for each time interval from $T_i$ to $T_{i+1}$, the bonus term for the first communication can always be trivially bounded by 1 for all agent $m$. Therefore, the summation of the regret over first communication can be upper bounded by 1. In addition, since the norm of each action $\|\mathbf{x}\|_2 \leq L$ and it implies that we have

$$\det(\boldsymbol{\Sigma}_T^{\text{all}}) \leq (\lambda + TL^2)^d, \tag{6.8}$$

which implies that the number of different intervals is at most $d\log(1 + TL^2/\lambda)$. Combining the upper bound of regret in (6.6), (6.7) and (6.8), we have

$$\text{Regret}(T) \leq dM\log(1 + TL^2/\lambda) + \sum_{t=1}^{T} 2\sqrt{2(1 + M\alpha)}\beta \|\mathbf{x}_{T_{i,2}}\|_{(\boldsymbol{\Sigma}_{T_{i,2}}^{\text{all}})^{-1}}$$
$$\leq d\log(1 + TL^2/\lambda) + 2\sqrt{2(1 + M\alpha)}\beta \sqrt{2dT\log(1 + TL^2/\lambda)},$$

where the last inequality follows from a standard elliptic potential argument (Abbasi-Yadkori et al., 2011). This completes the proof. $\square$

---

[5]There is no comminucation happening at $t_0$ or $t_N$, but we include them for notational convenience.

# 7 Experiments

In this section, we report numerical simulation results on the comparison between our algorithm with other baselines. Specifically, we construct a contextual linear bandit instance with feature dimension $d = 25$. In each round $t$, the active agent $m_t$ is uniformly sampled from all $M$ agents. We set the total number of rounds $T = 30,000$ and test for $M = 15$ and $30$. We compare our `FedLinUCB` with Async-LinUCB (Li and Wang, 2022a) and OFUL Abbasi-Yadkori et al. (2011) with full communication (i.e., the active agent communicates with the server in each round). Due to space limit, further details and more simulation results are deferred to Appendix B. The code and data for our experiments can be found on Github [6].

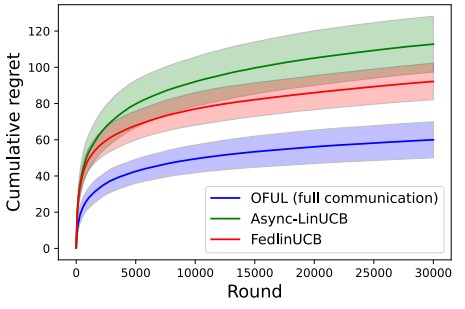
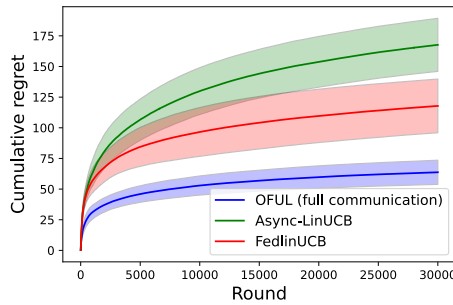

(a) $M = 15$. Cumulative regret versus Round      (b) $M = 30$. Cumulative regret versus Round

Figure 2: Plots of cumulative regret versus round for $M = 15$ (Fig. 2(a)) and $M = 30$ (Fig. 2(b)), comparing `FedLinUCB` (ours) with Async-LinUCB (Li and Wang, 2022a) and OFUL (Abbasi-Yadkori et al., 2011) with full communication. The results are averaged over 20 runs with the error bars chosen as the empirical one standard deviation.

It is clear from Figure 2 that `FedLinUCB` outperforms Async-LinUCB (Li and Wang, 2022a) in terms of regret. Although OFUL with full communication has the smallest regret, its communication cost $(2MT)$ is much higher than ours. Furthermore, we plot the log-scaled regret in Figure 3(c) and 3(d) in Appendix B, which show that the average regret of our algorithm actually has a rate very close to the optimal rate of OFUL. Overall, the numerical simulation corroborates our theoretical results.

# 8 Conclusion and Future Work

In this work, we study federated contextual linear bandit problem with fully asynchronous communication. We propose a simple and provably efficient algorithm named `FedLinUCB`. We prove that `FedLinUCB` obtains a near-optimal regret of order $\widetilde{O}(d\sqrt{T})$ with $\widetilde{O}(dM^2)$ communication complexity. We also prove a lower bound on the communication complexity, which suggests that an $\Omega(M)$ communication complexity is necessary for achieving a near-optimal regret. There still exists an $O(dM)$ gap between the upper and lower bounds for the communication complexity and we leave it as a future work to close this gap. Another important direction for future work is to study federated linear bandits with a decentralized communication network without a central server (i.e., P2P networks). In addition, there are potential privacy concerns when the agents upload and download data from the server, and it remains an open problem to devise provably efficient algorithm for asynchronous federated linear bandits with privacy guarantees.

## Acknowledgments and Disclosure of Funding

We thank the anonymous reviewers and area chair for their helpful comments. JH and QG are partially supported by the National Science Foundation CAREER Award 1906169 and the Sloan Research Fellowship. The views and conclusions contained in this paper are those of the authors and should not be interpreted as representing any funding agencies.

---

[6]https://github.com/uclaml/FedLinUCB

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
