# OpenReview forum: "A Simple and Provably Efficient Algorithm for Asynchronous Federated Contextual Linear Bandits"
_NeurIPS.cc/2022/Conference — NeurIPS 2022 Accept_

### Official Review · Reviewer_BhjP · 2022-07-10

**Rating:** 6
**Confidence:** 3
**Soundness:** 4 excellent
**Presentation:** 4 excellent
**Contribution:** 2 fair

**Summary:**

This paper proposes the FedLinUCB algorithm for asynchronous federated contextual linear bandits. The authors prove that the algorithm has a tight regret upper bound and low communication complexity.

**Questions:**

Why don't you provide simulation results? If my guess is right, in numerical simulation, FedLinUCB would have almost the same performance as [Li and Wang 2022], if not a little worse. But FedLinUCB could still have a smaller communication time.

**Limitations:**

This part of the paper looks good for me.

**Strengths And Weaknesses:**

1. Strengths
	1. Propose a better communication protocol that fits the asynchronous setting;
	2. Provide a new method to derive LinUCB's upper bound in the asynchronous setting.
	3. The author did a very good job in presenting their work.
2. Weaknesses:
	1. My major concern lies in the significance. The strengths above are not major compared to [Li and Wang 2022, AISTATS]. Because FedLinUCB is a (very) minor modification of [Li and Wang 2022] in communication protocol: given the same communication condition (determinant-based criterion), the difference is only in *how* agent and server communicate. (also see the paper's Table 1's last two rows).
	2. Secondly, in my opinion, the originality also does not reach the NeurIPS bar. The model is not new. The algorithmic technique is already broadly used (already in the same model as well). Only the analysis technique is novel, though seems minor (since it only works in this specific federated contextual linear bandits' LinUCB algorithm).

------
After rebuttal: I thank the author for the detail comparison [Li and Wang, 2022] and the additional experiments. My concerns are addressed. I raised my score to weak accept.

---

> ### Author Response · Authors · 2022-08-02
> **Response to Reviewer BhjP (part 2/2)**
>
> **Q2**: The originality of the paper is limited. In my opinion, the originality also does not reach the NeurIPS bar. The model is not new. The algorithmic technique is already broadly used (already in the same model as well). Only the analysis technique is novel, though seems minor (since it only works in this specific federated contextual linear bandits' LinUCB algorithm).
>
> **A2**: We agree that the contextual linear bandit model and some algorithmic designs (determinant-based criterion) are not new and have been studied in the existing literature. However, we would like to emphasize that the asynchronous federated linear bandits setting is new and of broader interest. As we have discussed in our paper, there are some existing efforts towards federated linear bandits. Yet, they are either limited to the synchronous setting [Wang et al., 2019; Dubey and Pentland, 2020; Huang et al., 2021] or not fully asynchronous with strong assumptions [Li and Wang, 2022] (whose proof is flawed as we discussed above).
>
> Our work provides the first asynchronous federated linear bandit algorithm that is simple, efficient, and provably correct. The analysis of our algorithm involves novel proof techniques by first establishing the local concentration of each agent’s data and then relating it to the global concentration of all data. These are critical to establishing the regret bound in the asynchronous setting. Also, we believe such proof techniques can be applied to other problems like asynchronous federated reinforcement learning.
>
> ---
>
> **Q3**: Why don't you provide simulation results? If my guess is right, in numerical simulation, FedLinUCB would have almost the same performance as [Li and Wang 2022], if not a little worse. But FedLinUCB could still have a smaller communication time.
>
> **A3**: Thanks for the suggestion! We have run simulations on synthetic data to compare our algorithm with the algorithm proposed by [Li and Wang, 2022] (although their analysis is flawed, their algorithm is still a practical algorithm). The settings and results are reported in Appendix A.3 in the rebuttal revision, and a brief summary of the experiment results can be found in the summary of rebuttal revision.

---

> ### Author Response · Authors · 2022-08-02
> **Response to Reviewer BhjP (part 1/2)**
>
> We would like to thank the reviewer for acknowledging the contributions and strengths of our paper and for the constructive suggestions and feedback. Below we provide clarifications and additional results in response to the questions and comments.
>
> **Q1**: The contribution of this work is not significant compared to [Li and Wang 2022, AISTATS].
>
> **A1**: We would like to emphasize that our contribution is very significant compared to [Li and Wang, 2022], which we clarify from the following three aspects:
> 1. **Flexible communication protocol.** Specifically, the communication protocol of  their proposed algorithm is more restricted than ours because in their algorithm uploading by one agent may trigger the other agents to download the latest data from the server. Even if those agents only wish to run the job locally and do not want to communicate, they will be forced to do so ( i.e., communicate with the server to fetch the data, update their policy accordingly, and then run the job). Such a communication protocol could be problematic since it completely neglects the common real-world scenario that the agents are offline or lose connection with the server, or sometimes the client simply does not want to communicate for some reason. As a comparison, in our Algorithm 1, the communication between the agent and the server (Line 9 and 12) involves only the participating agent, which is completely independent of other agents. This is clearly more flexible than the communication protocol in [Li and Wang, 2022] and does not suffer from the aforementioned issues.
> 2. **Mild assumptions for analysis.** [Li and Wang, 2022] imposed strong assumptions on the contexts in their Assumption 1. First, the distribution of the context vector needs to have a fixed covariance matrix across all time steps, i.e., $E_{t-1} [x_{t,a} x_{t,a}^\top ] = \Sigma_c$ for all $t$. Second, they require a minimum eigenvalue condition that $\Sigma_c \succcurlyeq \lambda_c$ for some $\lambda_c >0$. In fact, their assumption violates the standard setup of contextual linear bandits [Abbasi-Yadkori et al, 2011], where the context vectors can be arbitrary or even adversarial. In this sense, their algorithm is not an “authentic” contextual linear bandits algorithm.
> 3. **Sound theoretical analysis.** More importantly, the proof in [Li and Wang, 2022] is flawed and hard to fix. In detail, in the asynchronous setting,  we cannot define a fixed filtration as we do in the synchronous setting due to the asynchronous communication (as illustrated at the beginning of Section 6), which is caused by the fact that the data at the server do not have a fixed order. This further implies that in the asynchronous setting, the reward estimator based on the server-end data can be biased. Therefore, existing concentration results [Abbasi-Yakdori, 2011] cannot be directly applied, and hence the proof in [Li and Wang, 2022] is problematic. Please see Appendix A.1 for more details and for a counterexample. We have communicated with the authors of the paper about the flaws in their proof, and they acknowledged this flaw and were not able to give a sound fix.

---

> ### Author Response · Authors · 2022-08-04
> **Thank you for your quick and positive feedback!**
>
> We are glad that our responses and experiments have addressed your previous concerns.
>
> Thank you for raising the score!

---

### Official Review · Reviewer_eNSj · 2022-07-11

**Rating:** 6
**Confidence:** 3
**Soundness:** 3 good
**Presentation:** 3 good
**Contribution:** 2 fair

**Summary:**

The paper considers asynchronous federated contextual linear bandits problem, where each agent faces a single contextual linear bandits model and the parameters are the same. The learning objective is to minimize the cumulative regret of all agents. By a novel analysis, the paper prove that a simple algorithm called FedLinUCB enjoys regret upper bound $O(d\sqrt{MT})$, where d is the dimension of parameter space, M is the number of agents, and T is the time horizon. The paper also gives an lower bound of the communication cost scales $\Omega(dM)$.

**Questions:**

As mentioned above, IMHO, there may be some benefits if inactive agents are allowed to download new information. Is it possible to close the gap of communication cost upper and lower bound by personalized event-trigger downloading procedure?


------------
Thank the author for the explanation. My concerns are addressed and I decide to keep my score.

**Limitations:**

Yes

**Strengths And Weaknesses:**

The primary novelty of the paper is to provide an analysis for fully asynchronous federated linear contextual bandits, where each agent can independently decide whether it communicates with the central server. The paper is well-written and sound.

While the independency between agents' communication is new, the algorithm also restricts the inactive agents to download new information from the server.

---

> ### Author Response · Authors · 2022-08-02
> **Response to Reviewer eNSj**
>
> First we thank the reviewer for the effort, the positive support of our paper and the constructive feedback. Please see our response to the reviewer’s question below.
>
> **Q**: There may be some benefits if inactive agents are allowed to download new information. Is it possible to close the gap between the communication upper and lower bound by personalized event-trigger downloading procedure?
>
> **A**: First of all, we would like to make the following clarification. When we say an agent is “inactive”, we mean the agent is offline so it cannot upload or download data. If an agent can download data, we would consider it an “active” agent rather than an “inactive” agent in our setting.
>
> So we guess your question is about: what if an active agent would like to download data even though it does not upload data? Regarding this question, our algorithm can readily accommodate it if any agent wants to download new information from the server. However, since every single agent does not know the information of other agents and the data collected by the server unless the agent communicates with the server, it is not clear what criterion should be used to determine whether and when to download new information. In other words, the agent does not know if it would be significantly beneficial to download new information (which causes communication costs). Therefore, currently, we are not sure whether a personalized downloading procedure can help close the gap in the communication cost. It is definitely an important open problem for future work.
>
> Please let us know if we have misunderstood your question.

---

### Official Review · Reviewer_S7tZ · 2022-07-11

**Rating:** 5
**Confidence:** 4
**Soundness:** 3 good
**Presentation:** 3 good
**Contribution:** 2 fair

**Summary:**

The manuscript proposes a federated contextual linear bandit algorithm in the asynchronous setting, where all agents work independently and the communication between one agent and the server will not trigger other agents’ communication. The proposed algorithm obtains a near-optimal regret and low communication complexity.

**Questions:**

(a) The proposed algorithm shows that at each time $t$, only one client exchanges parameters to the server. Is this setting reasonable? Does this make the collaboration less effective? Are there any practical applications fit for this setting?

(b) How about the performance of the proposed algorithm in dealing with some real-world applications?


**Limitations:**

(a) In Algorithm 1, the parameters exchanged between the server and the client are $\Sigma$ and $u$, which may lead to the privacy leakage of local data. I suggest the authors refer to the work [Dubey and Pentland, 2020] to address this issue.

(b) It would be better to conduct some numerical experiments to support the merits of the proposed algorithm.

**Strengths And Weaknesses:**

Strengths

(a) This is the first asynchronous federated linear contextual bandit model considering the following features: (i) Each agent can decide whether or not to participate in each round; (ii) The communication between each agent and the server is asynchronous and totally independent of other agents.

(b) Considering that the order of the interaction between the agent and the environment is not fixed, standard martingale-based concentration inequality cannot be directly applied. This work applies a novel proof technique to solve this issue, which could be interesting.

(c) The proposed algorithm achieves a near-optimal regret, low communication complexity and switching cost simultaneously.

Weaknesses

(a) The proposed algorithm shows that at each time $t$, only one client exchanges parameters to the server. It seems that this setting is not reasonable and could make the collaboration less effective.

(b) In Algorithm 1, the parameters exchanged between the server and the client are $\Sigma$ and $u$, which may lead to the privacy leakage of local data.

(c) The motivation of this work is not very clear. The authors should pay more attention to this issue.

(d) There lack of numerical experiments to support the theoretical findings.

---

> ### Author Response · Authors · 2022-08-02
> **Response to Reviewer S7tZ (part 2/2)**
>
> **Q3**: The motivation of this work is not very clear.
>
> **A3**: We would like to clarify our motivation as follows:
> 1. Linear bandit is a simple yet powerful model for real-world applications, and the increasing amount of data being distributed requires a federated version of linear bandits.
> 2. However, most existing algorithms for federated linear bandits only work for the synchronous setting, and the asynchronous algorithm proposed by [Li and Wang, 2021] has limitations in their communication protocol and data assumption (their proof is also flawed, as we discuss in Appendix A).
> 3. Therefore, our goal is to develop a _fully_ asynchronous algorithm for federated linear bandits, and we propose a simple algorithm to achieve this goal.
> 4. Moreover, we identify and address unique theoretical challenges in a novel way, which we believe could be of independent interest in other asynchronous scenarios like federated reinforcement learning.
>
> We have revised our paper to further emphasize the above motivation. We would appreciate it if the reviewer can tell us which specific part needs further motivation.
>
> ---
>
> **Q4**: It would be better to conduct some numerical experiments to support the merits of the proposed algorithm. Also how about the performance of the proposed algorithm in dealing with some real-world applications?
>
> **A4**: Thanks for the suggestion! We did not perform experiments on real-world datasets, but we have added experiments on synthetic data. We added a new section in the appendix to describe the experiment setup and results, where we compared our algorithm with some baselines. Please see a brief summary of the experiment results in the summary of rebuttal revision, and also see Appendix A.3 in the rebuttal revision for the complete results.

---

> ### Author Response · Authors · 2022-08-02
> **Response to Reviewer S7tZ (part 1/2)**
>
> We would like to thank the reviewer for acknowledging the contributions and strengths of our paper and for the thoughtful feedback. Below are our responses to the questions and comments.
>
> **Q1**: The proposed algorithm shows that at each time t, only one client exchanges parameters to the server. Is this setting reasonable? Does this make the collaboration less effective? Are there any practical applications fit for this setting?
>
> **A1**: We believe there is some misunderstanding about our setting. We apologize if we did not make it clear. We would like to clarify that here $t \in [T]$ only denotes the index of the rounds, and it merely indicates the order of clients (i.e., agents) participating in the bandit problem, rather than the ‘real time’ of participation for the agents. In other words, if there is more than one client participating (e.g., exchanging parameters with the server) within a very short interval of time, there is still an order of occurrence among these participation events (i.e., even if the occurrence time of two close events only differ by milliseconds, there is an order), so the client participation will still happen in a sequential order that can be indexed by round $t$.
>
> In addition, our algorithm can be equivalently rewritten to reflect the application scenarios where a group of participation happens at the same round, as demonstrated by Algorithm 2 in Appendix A. Note that in Algorithm 2: we use round $k$ as the index instead of `time’ $t$ (line 2); we allow a group of participants in one round (line 3). The form of Algorithm 2 aligns with those of existing algorithms in, e.g., [1, 2].
>
> In fact, our setting is more general than existing federated (distributed) linear bandits work [1, 2],  because they require full participation of all the clients in each round (all $M$ clients need to be active in each round), but we allow partial participation (any subset of $M$ agents). Our setting is also more flexible than [3] because in our setting the communication between one client and the server will never trigger the communication between the server and other clients, but their setting [3] will.
>
> So our setting is actually very flexible and reasonable and fits practical applications very well. Our setting does not prevent collaboration among clients at all.
>
> _Reference_:
>
> [1]: Distributed bandit learning: Near-optimal regret with efficient communication, Yuanhao Wang, Jiachen Hu, Xiaoyu Chen, Liwei Wang, ICLR 2020.
>
> [2]: Differentially-private federated linear bandits, Abhimanyu Dubey, Alex Pentland, Neurips 2020.
>
> [3]: Asynchronous upper confidence bound algorithms for federated linear bandits, Chuanhao Li, Hongning Wang, AISTATS 2022.
>
> ---
>
> **Q2**: In Algorithm 1, the parameters exchanged between the server and the client are $\Sigma$ and $u$, which may lead to the privacy leakage of local data. I suggest the authors refer to the work [Dubey and Pentland, 2020] to address this issue.
>
> **A2**: Thanks for pointing this out and it is a great point! We think in principle such privacy concerns can be mitigated by properly ejecting noises to $\Sigma$ and $u$ following [Dubey and Pentland, 2020]. Nevertheless, the main focus of our work is to develop a fully asynchronous federated linear bandits algorithm with provable guarantees, which itself is a significant contribution and can serve as a solid first step towards devising differentially private asynchronous federated bandit algorithms. We have added a discussion in the conclusion and future work section regarding this important issue.

---

> ### Author Response · Authors · 2022-08-07
> **Follow up with Reviewer S7tZ**
>
> Dear Reviewer,
>
> Since the deadline for the author-reviewer discussion phase is fast approaching, we would like to follow up with you to see if you have any further questions.
>
> In our rebuttal, we have addressed all your questions. In particular, per your suggestion, we have added numerical experiments  to corroborate our theory and compare our algorithm with several baselines. We are looking forward to your feedback. Thank you.
>
> Best,
>
> Authors of Paper 4870

---

### Author Response · Authors · 2022-08-02
**Summary of rebuttal revision**

First, we would like to thank all the reviewers for their careful reading and insightful comments and questions!

In complement to our response to the reviewers, we have uploaded a revised paper. Besides correcting some minor typos, some major changes are marked out by blue in the PDF, and we summarize the major updates as follows:
1. We revise the writing of the introduction section to better motivate our work.
2. We updated Table 1 for a clearer comparison among different algorithms.
3. We have revised the writing of the comparison between our results and those of [Li and Wang, 2022] in Appendix A.1. This also corresponds to Reviewer BhjP’s comments on the comparison between our work and [Li and Wang, 2022].
4. We have conducted numerical experiments to compare our algorithm with other baselines. The simulation results reported in Appendix A.3 corroborate our theory. This addresses the questions on the simulation of the proposed algorithm by Reviewer S7tZ and Reviewer BhjP. A brief summary of the experiment is provided below.

**Summary of the experiments**:

_Experiment setup_: In detail, we construct a linear bandit instance with dimension d=25 and true model parameter $\theta^*=[1/\sqrt{d},\ldots,1/\sqrt{d}]$. In each round $t \in [T]$, an active agent $m_t$ is uniformly sampled from $M$ agents and the decision set $\mathcal{D}_t$ consists of 25 different actions, where each action is uniformly sampled from $[-1/\sqrt{d},1/\sqrt{d}]^d$. After choosing the action, the active agent will receive the reward perturbed by a Gaussian noise (R=0.3). We run a simulation on the above linear bandit instance with a total number of rounds T=30000 (averaged over 20 runs) and the number of agents is set to be $M=15$ or  $M=30$. We implement our FedLinUCB algorithm and compare its performance with Async-LinUCB [Li and Wang, 2022] and OFUL [Abbasi-Yadkori et al., 2011] with full communication (i.e., the active agent communicates with the server during each round). We tune the parameter $\alpha$ for FedLinUCB and $\gamma_U=\gamma_D$ for Async-LinUCB to ensure a fair comparison. We found that $\alpha=1$ for FedLinUCB and $\gamma_U=\gamma_D=5$ for Async-LinUCB lead to the best performance for both algorithms.


_Results_: Here we report part of the simulation results with $M=15$ agents.

| Algorithm\ Cumulative regret \Rounds |   5000    |  10000  |  15000 |     20000   | 25000       | 30000    |
|:-------------:    |:---------: |:--------:|:--------:|:---------:   |:------:       |:--------:  |
|OFUL with full communication   |      42.5     |  49.4    | 53.3    | 56.0    | 59.1 | 59.9 |
| Async-LinUCB  ($\gamma_U=\gamma_D=5$)                 |   79.7     | 92.0   |  99.6     |105.0     |109.2  |112.7 |
|FedLinUCB ($\alpha=1$)                              |     67.8       |  76.9    |  82.1  |  85.9   |  89.2  | 92.1|

| Algorithm\ Communication cost \Rounds |   5000    |  10000  |  15000 |     20000   | 25000       | 30000    |
|:-------------:    |:---------: |:--------:|:--------:|:---------:   |:------:       |:--------:  |
|OFUL with full communication   |      10000     |  20000    | 30000    | 40000   | 50000 | 60000 |
| Async-LinUCB ($\gamma_U=\gamma_D=5$)                        |   487.5       |  570.0   |  621.75   |660.0     |688.5  |705.0 |
|FedLinUCB  ($\alpha=1$)                            |    284.8       |  329.1    |  354.9  |   373.9   |  388.1  | 400.4|

These simulation results suggest that our FedLinUCB algorithm significantly outperforms the Async-LinUCB algorithm proposed by [Li and Wang, 2022], as our algorithm achieves lower regret with lower communication cost.



Thank you,

Authors

---

### Meta-Review · Area_Chair_8rcD · 2022-09-06

**Recommendation:** Accept
**Confidence:** Certain

**Metareview:**

The reviewers all recommend acceptance (with various extent), and the AC also shares their opinion.

Regarding the experiments, please make sure that the final version of the paper complies with the relevant parts of the checklist (e.g., report error bars). It could also be interesting to see an experiment where $\theta^*$ is non-uniform.

**Award:**

No

---

### Decision · Program_Chairs · 2022-09-14

Accept